# Evidence of Validity and Measurement Invariance by Gender of the Vaccination Attitudes Examination (VAX) Scale in Colombian University Students

**DOI:** 10.3390/jcm11164682

**Published:** 2022-08-10

**Authors:** Begoña Espejo, Marta Martín-Carbonell, Kelly Carolina Romero-Acosta, Martha Fernández-Daza, Yadid Paternina

**Affiliations:** 1Department of Methodology of Behavioral Sciences, University of Valencia, Av. Blasco Ibáñez, 21, 46010 Valencia, Spain; 2Psychology Department, Cooperative University of Colombia, Troncal del Caribe S/N, Santa Marta 470002, Colombia; 3Faculty of Humanities and Education, Corporación Universitaria del Caribe (CECAR), Sincelejo 700001, Colombia

**Keywords:** vaccine hesitancy, vaccine refusal, confirmatory factor analysis, structural equation modeling, measurement invariance, concurrent validity, psychometric properties, COVID-19

## Abstract

Background: Having a valid tool to assess attitudes toward vaccination and identify the concerns that drive vaccine refusal can facilitate population studies and help guide public health interventions. The objective of this study has been to adapt the Vaccination Attitudes Examination (VAX) scale in Colombian university students and to study its psychometric properties in a non-probabilistic sample of 1074 Colombian university students. Methods: A confirmatory factor analysis was used to study the factorial structure. A structural equation model was tested to study concurrent validity and to check whether the factors predicted having received the coronavirus vaccine. Gender-based measurement invariance was also studied for the best model. Results: The results support the structure of four related factors. The composite reliability index was good for all the factors, but the average variance extracted was not as good for the second factor. There was strong measurement invariance by gender, and two factors are good predictors of being vaccinated or not. Conclusions: The VAX has shown construct and concurrent validity and is a reliable tool for evaluating attitudes towards vaccines in university students in Colombia. It may help guide the implementation of actions for the National Vaccination Plan and institutional policies.

## 1. Introduction

Despite the demonstrated efficacy of vaccines in reducing mortality and morbidity from communicable diseases, vaccination rates are declining in many areas of the world [1,2], even before the pandemic caused by the SARS-CoV-2 virus became the main topic of newspapers and the subject of passionate debates on social networks and even within families [3,4]. The decision not to be vaccinated is influenced by different reasons, such as forgetfulness or lack of time, concerns about medical interventions that are considered “unnatural”, specific concerns about the safety of vaccines, or mistrust of the motivations of pharmaceutical companies and governments to promote vaccines [2,5]. It may also be the case that, with increasing vaccination rates, some people selfishly rely on the indirect protection provided by others’ vaccinations and thus avoid the effort and risk of possible adverse effects [6].

In line with these issues, and according to the World Health Organization [2], there are some types of events that decrease confidence in vaccines. For example, reactions to vaccines, critical reports in the media, and possible confusion between vaccines, among other reasons. This organization also indicates that the presence of several factors can negatively interfere with the decision to be vaccinated. For example, a pandemic outbreak situation, a mass immunization campaign, or the introduction of a new vaccine. In addition, these latter two factors are usually mandatory in a pandemic situation. Thus, today, with the coronavirus pandemic, there are many people who are wondering if there are sufficient reasons to consider it convenient to get vaccinated [2].

The diversity of predictive psychosocial factors of the acceptance of vaccination against COVID-19 has been pointed out, such as political opinions, attitudes towards science, antisocial tendencies [7], or prosociality [8]. An international study that included 24 countries and more than 5000 participants found that anti-vaccine attitudes were more frequent in people who had a high level of conspiratorial thinking. Hierarchical worldviews and strong individualism were also observed, as well as high levels of displeasure for blood and needles [9].

In Colombia, historical deficits in vaccination are recognized, which have been fundamentally explained by inequities in access to health services [10], although there are also some studies on antecedents of distrust and rejection of vaccination. In a very recent study, the feelings expressed by Colombian Twitter users towards vaccination against COVID-19 were analyzed, finding a predominance of negative feelings towards vaccines [11]. In addition, another recent study found distrust and reluctance to vaccination against COVID-19 in Colombians over 80 years of age [12]. On the other hand, another study found a prevalence of negative attitudes toward COVID-19 vaccines in university students [13].

In relation to other health problems, evidence has been found that the negative attitudes of Colombian health professionals were the main cause of missed opportunities for vaccination of young children [14]. It is worth highlighting what happened with the human papillomavirus (HPV) vaccine in June 2014, when adverse effects such as headache, paresthesia, shortness of breath, chest pain, and fainting were reported in more than 500 girls residing in a municipality called Carmen de Bolívar. This is a small town in the north of Colombia which faces great socioeconomic difficulties and is severely affected by domestic and political violence. The investigation concluded that the symptoms reported by the girls had no biological relationship to the vaccine and could be explained by a mass stress reaction. However, this issue was sensationalized by the national media and social media, causing public confidence in the HPV vaccine to plummet. For this reason, national coverage rates decreased from more than 80% in 2014, to 14% and 5% for the first and second dose, respectively, in 2016 [15,16]. In fact, it has also been found that there is a strong lack of knowledge and a severe distrust of the HPV vaccine in health professionals [17].

Thus, understanding the attitudes that underlie vaccination reluctance is important for the development of effective public health interventions [18]. Being able to have a valid tool to assess attitudes towards vaccination, and identify the concerns that drive vaccine refusal, can facilitate population studies and help guide public health interventions [19]. In this regard, numerous scales have been developed to assess attitudes toward vaccines, but these often focus on particular populations or specific vaccines [5,20,21]. Methodological limitations have also been found in the measures available for the evaluation of specific problems, such as attitudes towards papillomavirus (HPV) vaccines [22] or the scales developed to assess the confidence of parents towards childhood vaccination [23]. On the other hand, although it is true that attitudes towards vaccination vary according to the type of vaccine and the health problem [24], there are reasons to believe that a single measure can be an efficient way of identifying people with concerns related to vaccines in general [5]. A review of scales to study attitudes towards childhood vaccination reported that most shared indicators about beliefs about the benefits or importance of vaccination, confidence in vaccines and in health care providers, confidence in the legitimacy of authorities to require vaccination, vaccination harms, and perceived risks of infectious diseases [23].

The Vaccine Attitudes Examination (VAX) scale [5] was developed in order to generate a multifaceted tool to assess general attitudes towards vaccines. The authors developed an initial pool of 45 items extracted from focus groups with people who identified themselves as being favorable to vaccines and with people who distrusted vaccines. They also reviewed anti-vaccine literature and internet sites. To delimit the subscales, they relied on exploratory and confirmatory factor analysis techniques, managing to reduce the scale to 12 items that could be grouped into four related factors. Validation studies have been carried out in the United Kingdom [25], Romania [26], Turkey [27], and Spain [28]. In all of them, the structure of four factors described in the original study has been confirmed. Evidence of its validity in predicting vaccination refusal, both in oneself and in one’s children, has also been reported [5,25].

The objective of this research was to obtain evidence of the validity of the VAX to study attitudes towards vaccination in Colombian university students. This population was chosen because university students can have leadership in their families and communities since they are active young people in different contexts of life and can come to exert an important influence on their families and the community. In this sense, they can influence the acceptance of vaccines [29] since the data support the possibility that the opinions in favor of vaccination of friends and relatives can be exploited to reduce vaccine hesitancy [30]. However, on the other hand, some studies have shown that university students could have negative attitudes towards vaccination [31,32].

Specifically, in this project we were interested in evaluating whether the factorial structure of the VAX described in other populations is confirmed, taking into account that in some studies it has been proposed that the construct “attitudes towards vaccination” could be one-dimensional [33]. In addition, although the VAX adaptations carried out in various populations have corroborated the structure of four related factors [25,26,27], in the Spanish adaptation, the factorial structure is not clear [28]. We also investigated whether there is gender invariance since we did not find any study that refers to it. Finally, information has been obtained about the predictive validity of the VAX with respect to the behavior of being vaccinated or not vaccinated against COVID-19.

## 2. Materials and Methods

### 2.1. Procedure

The data comes from an online survey that includes other instruments within the framework of a broader study which aims to investigate well-being in Colombian university students and its relationship with personality variables and experiences with COVID-19. This study, in turn, is part of an ongoing project led by the Research Initiatives Working Group (RIWG) at the American Psychological Association (APA) Interdivisional Task Force on the Pandemic [34]. The project is entitled “International and Multidimensional Perspectives on the Impact of COVID-19 (IMPACT-C19)” and comes with data and multidisciplinary expertise involving over 150 members across major geographical regions in nearly 80 countries [35].

The data were collected between August 2021 and May 2022 using the LimeSurvey platform, installed on the university’s servers. This allows us to guarantee the protection of the data by our university, in such a way that it is guaranteed that only the researchers can have access to the data. The survey was completely anonymous and voluntary. The link to it was sent via email and distributed on social networks, following the snowball process. Before starting the survey, the study was briefly explained and then participants were required to accept informed consent in order to begin responding.

### 2.2. Participants

A non-probabilistic sample of 1197 Colombian university students was obtained. The average age of the sample was 22.5 years (SD = 4.76, minimum 18, maximum 57). Table 1 summarizes the information on the sociodemographic variables. As can be seen, women predominated, as well as students of medium socioeconomic status, people from the departments of Magdalena and Sucre, and people who were studying only remotely when the information was obtained.

### 2.3. Variables and Instruments

The Vaccination Attitudes Examination (VAX) scale: This scale assesses anti-vaccine attitudes. The items are grouped into four dimensions: (1) trust of vaccine benefit; (2) concern about unforeseen future effects; (3) concern about commercial effects and speculation; and (4) preference for natural immunity. The measurement is carried out at six levels of the Likert scale (1—strongly disagree to 6—strongly agree), but there are studies that report its use with 5 response options [36].

For the present investigation, the Spanish adaptation developed for the International and Multidimensional Perspectives on the Impact of COVID-19 across Generations (IMPACT-C19) project was used. This adaptation was made according to the recommendations of the International Test Commission [37]. The version we used has five response options on a Likert scale ranging from 1—strongly disagree to 5—strongly agree, since the response options were reduced to unify the alternatives in all the questionnaires used in the study. 

IMPACT-C19 project: The translation was evaluated by 26 psychology students from a Colombian university in order to detect if any item or term was difficult to understand or caused discomfort. The result showed that 100% of the interviewed students evaluated that the test was easy to understand and answer, and that it did not cause discomfort. Items are shown in Appendix A. The rating for each factor is obtained by adding the scores for the questions on each subscale. Higher scores on factors 2, 3, and 4 reflect stronger anti-vaccine attitudes, while higher scores on factor 1 indicate positive attitudes toward vaccines. The scores on each subscale have theoretical scores between 5 and 15 since each subscale consists of 3 items.

To assess whether the participants were vaccinated, a question with a dichotomous response (1. Yes, 2. No) was asked: “Have you been vaccinated against COVID-19?”. Participants were also asked if they had completed the two-dose schedule.

### 2.4. Data Analysis

Confirmatory factor analysis (CFA) was used to study the factorial structure of the VAX. Two CFAs were calculated: a single-factor and a four-related factor. The maximum likelihood robust (MLR) estimator was used because some studies suggest that MLR estimation can be used in confirmatory models when the data distribution is not normal and if the number of response categories for items is greater than four [38,39]. In this case, the variability in the parameter estimates is relatively small and MLR offers less biased standard error estimates as well as good estimates of the correlations between the factors [40]. In addition to χ^2^, different indices have been used to determine model fit: the comparative fit index (CFI), the root mean square error of approximation (RMSEA), and the standardized root mean square residual (SRMR). These fit indices may each be influenced by numerous factors, such as sample size, data distribution, and model complexity and specifications. Therefore, we used both liberal and conservative cut-off points for acceptable fit for the CFI, RMSEA, and SRMR: the CFI should be close to or greater than 0.90 (liberal) or 0.95 (conservative), RMSEA should be 0.10 or less (liberal) or 0.06 or less (conservative), and SRMR should be less than 0.10 (liberal) or 0.05 (conservative) [41]. The factor measurement reliability was evaluated with the composite reliability index (CR) and the average variance extracted index (AVE). Values above 0.70 for both indices are considered good, and values above 0.50 for the AVE are considered acceptable [42]. For the model that best fitted the data, the corrected item-total polyserial correlations for the items in each subscale have been calculated [43] as indicators of corrected homogeneity indices for items with ordinal response scales [40].

Gender-based measurement invariance was also studied for the best model, evaluated by calculating three nested invariance models: configural, metric, and scalar. To assess the degree of invariance among the models, the usual cut-off points in the increase in the indices have been considered: a change of 0.010 or greater in CFI along with a change of 0.015 or greater in RMSEA, or a change of 0.030 or greater in SRMR would indicate that there is no invariance [44]. To study the concurrent validity of the scale, a structural equation model has been specified considering the best model for the VAX scale as predictor of vaccination. Since the outcome variable is dichotomous (vaccination, yes or no), the odds ratio of the logistic regression were also obtained. Furthermore, this validity model offers the estimation of the location parameter for the dichotomous variable (the parameter for the Rasch model). This parameter reports the minimum level of the trait from which a person is more likely to be vaccinated. Finally, descriptive statistics were obtained for each factor to propose normative values that guide the interpretation of individual scores.

CFA, corrected item-total polyserial correlations, measurement invariance, and concurrent validity analyses were carried out using Mplus 8.8 [45], and for the description of the sociodemographic variables and the statistics for the items of the VAX scale, the factors, and the normative values, IBM SPSS 26 (IBM, Armonk, NY, USA)was used.

## 3. Results

Regarding the construct validity, the one-dimensional CFA model was clearly inappropriate: χ^2^ (54) = 1946.10, *p* < 0.001, CFI = 0.473, RMSEA = 0.171, RMSEA 90% CI = [0.165, 0.178], and SRMR = 0.148. The four-factor model showed good fit: χ^2^ (48) = 250.55, *p* < 0.001, CFI = 0.944, RMSEA = 0.059, RMSEA 90% CI = [0.052, 0.067], and SRMR = 0.053. All factor loadings were statistically significant (*p* < 0.001) ranging from 0.485 to 0.906. This factor model can be observed in Figure 1 where statistically significant correlations were observed among the four factors. As can be seen, factor 1 had negative correlations with the other 3 factors, being the lowest with factor 4.

Regarding reliability, the composite reliability index (CR) was good for Factor 1 (CR = 0.880), Factor 2 (CR = 0.671), Factor 3 (CR = 0.802), and Factor 4 (CR = 0.794). The average variance extracted (AVE) was good for F1 (AVE = 0.710), F3 (AVE = 0.575), and F4 (AVE = 0.563), but less favorable for the second factor (AVE = 0.412). Table 2 shows the descriptive data of the VAX scale items and the corrected item-total corrected polyserial correlations. As can be seen, the values of the corrected homogeneity indices are adequate for all the items on the scale.

In Table 3 shows the results for the measurement invariance models by gender. The results indicate good fit of the four-factor model for women and acceptable fit for men. The results of the invariance model by gender showed strong invariance, therefore, means may be compared by gender. After fixing latent mean values to zero for men, no differences for gender were observed in any of the factors: (Factor 1) b = −0.054, z = −0.966, *p* = 0.334; (Factor 2) b = 0.003, z = 0.091, *p* = 0.928; (Factor 3) b = −0.053, z = −1.052, *p* = 0.293; and (Factor 4) b = −0.081, z = −1.824, *p* = 0.068).

Figure 2 shows the validity model considering the four-factor structural model as a predictor of vaccination. The results showed that the first two factors (F1 = trust of vaccine benefit and F2 = worries over unforeseen future effects) were good predictors of being vaccinated or not, respectively. The first factor was a positive predictor of vaccination (*p* < 0.001), and the second one was a negative predictor (*p* < 0.001). The coefficients of the other two factors were not statistically significant.

The odds ratio for the four factors are 1.911, 0.297, 1.420, and 0.969, respectively. Likewise, this validity model offers the estimation of the location parameter (b) for the dichotomous variable. In this case, the estimated value of b for the Vaccine variable is −0.563 (z = −15.360, *p* < 0.001). This means that, from a medium-low level on the trait, people are more likely to be vaccinated than not vaccinated. In case the model could show different results in its predictive capacity depending on whether the participants had been infected or not, we estimated the validity model with the sample of people who say they have not been infected or are not sure about it. The biggest difference is that statistically significant correlations are observed between all the factors, something that does not occur with the validity model in Figure 2. In this model (with all the sample), the correlation between factors 1 (trust of vaccine benefit) and 2 (worries over unforeseen future effects), and the correlation between factors 1 (trust of vaccine benefit) and 4 (preference for natural immunity) were not significant. However, when performing the analysis after eliminating the infected people, the correlation between these factors is significant and inverse. Despite this, this result does not bring about major changes in the validity model in which factor 1 (trust of vaccine benefit) and factor 2 (worries over unforeseen future effects) also continue to be significant predictors, and the odds ratio are also very similar. Again, we reduced the sample and estimated the model only with the people who say they have not been infected, and again the prediction model gives the same results as the one in Figure 2, except for the fact that only the correlation between factors 1 (trust of vaccine benefit) and 4 (preference for natural immunity) is not significant. However, factors 1 (trust of vaccine benefit) and 2 (worries over unforeseen future effects) also predict vaccination or not, and the odds ratio are very similar.

Finally, Table 4 shows the descriptive statistics of the VAX subscales and the provisional normative values.

## 4. Discussion

In the 21st century, it is being considered that the concept of validity must be supported by different sources of evidence. In the case of attitudes towards vaccines, the importance of terminological precision has been highlighted in order to understand where the problem lies, accurately measure it, and develop the appropriate interventions [46]. This study contributes to the evidence by establishing the construct validity and the concurrent validity of the VAX. This is a requisite to establish its ability to predict whether a person accepts vaccination or not

Regarding the factor structure, our results coincide with what was reported in the original study [5] and in the adaptations carried out in countries with different cultures, such as Romania [26] and the United Kingdom [25], in which a structure of four related factors has also been confirmed. These factors have also been commonly described in other measures of attitudes towards vaccines [20,22,23]. Likewise, the correlations between the factors are all statistically significant and in the expected direction. This result is consistent with the idea that attitudes towards vaccines can be considered a complex concept, as has been recognized in other studies [21].

We also found adequate reliability values for all subscales. An important point to note is that, as of 1999, the reliability of a test is now understood as a criterion of validity. In this sense, the values found for the CR are appropriate, although the AVE of the second factor is slightly low (0.40). However, some authors question the relevance of establishing fixed cut-off points to assess the AVE due to its variability depending on the number of instrument items and factor loadings [47]. Since the VAX factors are made up of only three items, it would be reasonable to state that, although the value is lower than the established cut-off point (0.50), the proportion of variance explained by the factor is adequate.

To our knowledge, this is the first study to examine the gender measurement invariance of the VAX. There are also few studies that investigate the measurement invariance by gender of other instruments to assess attitudes towards vaccination for specific health problems [48] or in specific populations [23]. However, it is important to ensure that the measurement instruments offer reliable information for both genders since the lack of scalar invariance would mean that men and women are thinking of different concepts when we measure attitudes towards vaccination. This would lead to a validity problem that would not allow a comparison between the two groups nor that both groups be studied together. Our results support the use of the VAX to study attitudes towards vaccines in female and male Colombian university students. Furthermore, no gender differences were found in any of the VAX factors.

Our study also demonstrated the concurrent validity of the VAX using structural equation modeling. Specifically, the subscales “Confidence in vaccines” and “Concern about adverse effects” were shown to be good predictors for predicting non-vaccination behavior against COVID-19 in Colombian university students. The finding that, in Colombian university students, confidence in the protective value of the vaccine (factor 1) predicts the behavior of getting vaccinated, while fear of adverse effects (factor 2) predicts the opposite, guides how interventions should be focused to promote vaccination in this type of population. It is also observed in the validity model that, at least in this sample, distrust of the government and pharmaceutical companies (factor 3) and the preference for natural immunity (factor 4) did not influence the decision to be vaccinated, unlike what has been reported in studies from other countries [49]. Therefore, communication strategies should focus on increasing confidence in the protective value of vaccines and calming concerns about possible adverse effects because these issues have been exaggerated by fake news, as well as by inappropriate messages distributed by the media [2].

In this sense, the main contribution of this study has been to make available to the Colombian academic community, researchers, and decision makers, a valid and reliable scale for the evaluation of attitudes towards vaccines in general, which also offers reference values for the evaluation of individual differences. This may help fine-tune the communication actions necessary for the successful implementation of vaccination plans.

### Limitations

One of the limitations of this research has been the impossibility of obtaining a national representative sample. We have had to resort to sampling by availability, which limits the generalization of the results, especially considering the cultural diversity of Colombia. Therefore, it would be very appropriate to study the psychometric properties of the VAX in other types of Colombian samples with different characteristics from the current ones, not only in university students. In addition, the administration of the online test reduced the opportunity for participation to only people with internet access. On the other hand, it would also be useful to consider people who do not believe in COVID and study their attitudes, not only towards the coronavirus vaccine, but towards vaccines in general. Another limitation is that the predictive validity was studied only with respect to the COVID-19 vaccine, so it would be important to investigate it for other types of vaccines. For these reasons, we recommend that the normative values provided be used with caution and for research purposes only.

In this investigation we did not examine either the temporal stability or the sensitivity of VAX to detect changes in attitudes towards vaccines; therefore, for future studies it is recommended to obtain test-retest comparisons. It would also be advisable to extend the validity study to other populations with a diversity of ages, educational levels, and occupations. Likewise, it will be enlightening to carry out an invariance analysis of other demographic variables, as well as to investigate whether the same factors predict whether or not to be vaccinated in other countries.

## 5. Conclusions

The psychometric properties of VAX have been studied using structural equation models. Likewise, the measurement invariance by gender has been analyzed. The results show that it is a valid and reliable tool for the evaluation of attitudes towards vaccines in university students in Colombia. Furthermore, it has shown its predictive value in terms of its ability to predict vaccination against the coronavirus. Using this scale to assess attitudes towards other vaccines may provide relevant information that helps guide more appropriate implementation of the actions of the National Vaccination Plan and institutional policies.

## Figures and Tables

**Figure 1 jcm-11-04682-f001:**
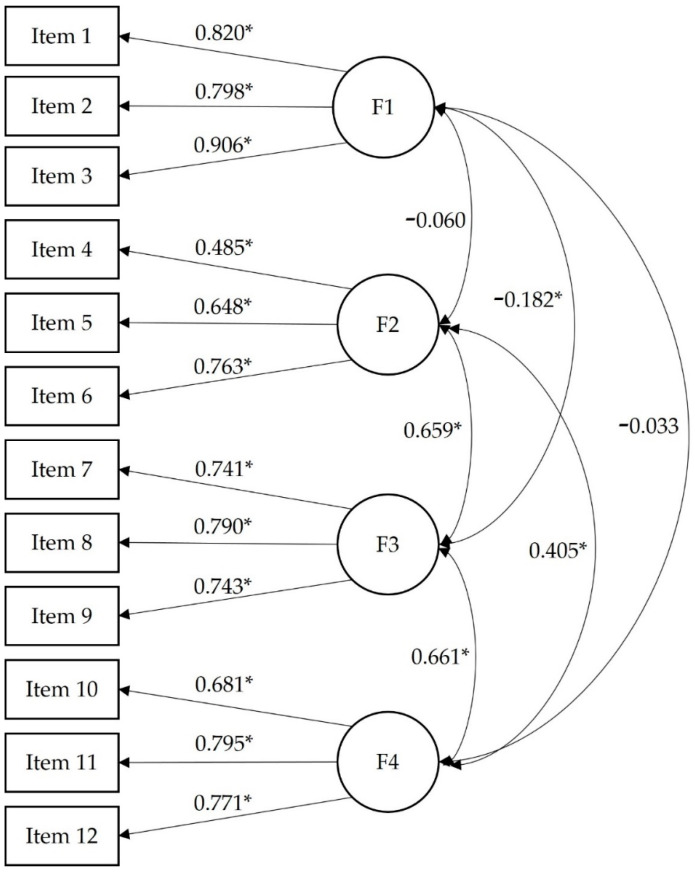
Standardized coefficients for the four-factor model of the Vaccination Attitudes Examination scale. Note: F1 = trust of vaccine benefit; F2 = worries over unforeseen future effects; F3 = concerns about commercial profiteering; and F4 = preference for natural immunity. * *p* < 0.001.

**Figure 2 jcm-11-04682-f002:**
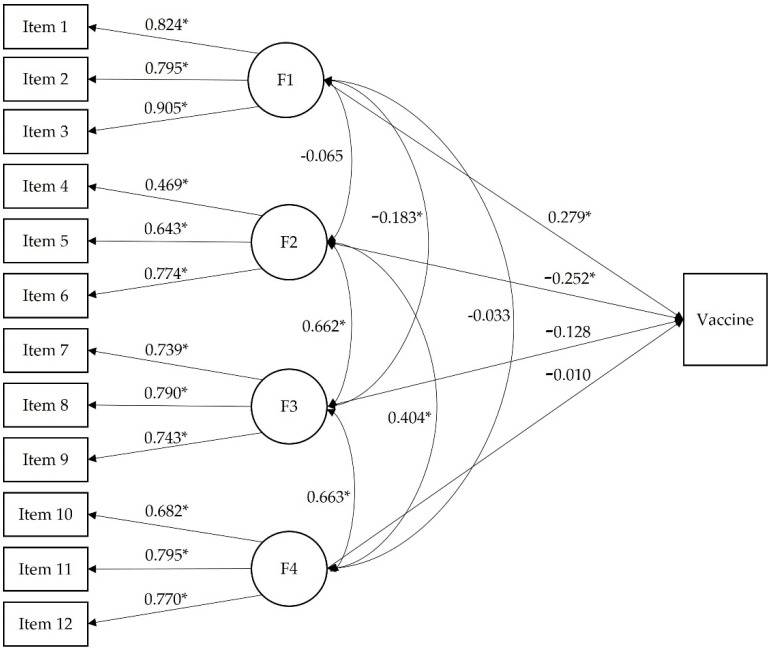
Standardized coefficients for validity model of the Vaccination Attitudes Examination Scale. Reference group for Vaccine: Yes. Note: F1 = trust of vaccine benefit; F2 = worries over unforeseen future effects; F3 = concerns about commercial profiteering; and F4 = preference for natural immunity. * *p* < 0.001.

**Table 1 jcm-11-04682-t001:** Frequencies and percentages of sociodemographic variables (N = 1197).

		n	%
Gender	Male	388	32.4
	Female	791	66.8
	Non-binary	3	0.3
	Prefer not to answer	7	0.6
Student condition	Full-time student	707	59.1
	Work and study	490	40.9
Socioeconomic level	High	9	0.8
	Medium-high	93	7.8
	Medium	435	36.3
	Medium-low	412	34.4
	Low	248	20.7
Provincial department where the studies were carried out	Amazonas	1	0.1
	Antioquia	80	6.7
	Arauca	2	0.2
	Atlántico	61	5.1
	Bolívar	14	1.2
	Boyacá	3	0.3
	Casanare	1	0.1
	Cauca	12	1.0
	César	22	1.8
	Córdoba	34	2.8
	Guainía	1	0.1
	La Guajira	13	1.1
	Magdalena	572	47.8
	Meta	4	0.3
	Norte de Santander	18	1.5
	Santander	16	1.3
	Sucre	338	28.2
	Tolima	1	0.1
	Valle del Cauca	4	0.3
Degree being studied	Psychology	339	28.3
	Social work	82	6.9
	Architecture	149	12.4
	Medicine	22	1.8
	Law	79	6.6
	Nursing	52	4.3
	Engineering	88	7.4
	Veterinary	2	0.2
	Other	384	32.1
How the participants were conducting their studies	Online	624	52.1
	Face-to-face	219	18.3
	Combined (online/face-to-face)	354	29.6
Participants infected with coronavirus	Yes	353	29.5
	No	555	46.4
	Not sure	246	20.6
	Missing values	43	3.6
Participants vaccinated against the coronavirus	Yes	874	73.0
	No	323	27.0

**Table 2 jcm-11-04682-t002:** Descriptive statistics and corrected item-total polyserial correlations for the items of the Vaccination Attitudes Examination Scale.

	Mean	Standard Deviation	Skewness	Kurtosis	Corrected Item-Total Polyserial Correlations	SE for the Corrected Item-Total Polyserial Correlations
Item 1	3.57	1.01	−0.59	0.14	0.779	0.006
Item 2	3.65	0.99	−0.67	0.26	0.763	0.007
Item 3	3.54	0.95	−0.46	0.11	0.829	0.005
Item 4	3.71	0.85	−0.71	0.83	0.441	0.018
Item 5	3.16	0.83	−0.11	0.40	0.515	0.016
Item 6	3.43	1.02	−0.42	−0.26	0.569	0.014
Item 7	3.21	0.95	−0.12	−0.08	0.656	0.011
Item 8	3.02	0.99	0.01	−0.18	0.725	0.008
Item 9	2.69	0.95	0.20	0.13	0.642	0.011
Item 10	2.91	0.91	0.04	0.23	0.593	0.012
Item 11	2.73	0.96	0.10	−0.11	0.706	0.009
Item 12	2.75	0.95	0.05	−0.07	0.668	0.009

Note: SE = standard error.

**Table 3 jcm-11-04682-t003:** Measurement invariance by gender models and goodness-of-fit indices. Reference group: Men.

	χ^2^	df	Δχ^2^	Δdf	CFI	RMSEA	SRMR	ΔCFI	ΔRMSEA	ΔSRMR
Men	168.71 *	48			0.909	0.081	0.070			
Women	154.26 *	48			0.954	0.053	0.048			
Configural	322.98 *	96	-	-	0.937	0.063	0.056	-	-	-
Metric	332.91 *	104	10.6	8	0.936	0.061	0.060	−0.001	−0.002	0.004
Scalar	356.93 *	112	20.0	8	0.932	0.061	0.062	−0.004	0.000	0.002

Note: Δχ^2^ = chi-square change; Δdf = degrees of freedom change; CFI = comparative fit index; RMSEA = root mean square error of approximation; SRMR = standardized root mean square residual; ΔCFI = CFI change; ΔRMSEA = RMSEA change; and ΔSRMR = SRMR change. * *p* < 0.001

**Table 4 jcm-11-04682-t004:** Descriptive statistics and percentiles of the subscales of the Vaccination Attitudes Examination Scale.

		F1	F2	F3	F4
N	Valid	1197	1197	1196	1197
	Missing	0	0	1	0
Mean		10.70	10.76	10.30	8.91
Median		11	11	10	9
Mode		12	12	9	9
Standard deviation		2.70	2.64	2.10	2.44
Variance		7.00	6.99	4.41	5.97
Minimum		3	3	3	3
Maximum		15	15	15	15
Percentiles	5	6	7	5	4
	10	7	8	6	6
	15	9	8	6	6
	20	9	9	7	6
	25	9	9	7	7
	30	9	9	8	7
	35	10	10	8	8
	40	10	10	9	8
	45	11	10	9	9
	50	11	10	9	9
	55	12	11	9	9
	60	12	11	9	9
	65	12	11	10	9
	70	12	11	10	9
	75	12	12	10	9
	80	12	12	11	10
	85	13	12	11	10
	90	15	13	12	11
	95	15	14	13	12

Note: F1 = trust of vaccine benefit; F2 = worries over unforeseen future effects; F3 = concerns about commercial profiteering; and F4 = preference for natural immunity.

## Data Availability

Not applicable.

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
