# Peer review of "Evidence of Validity and Measurement Invariance by Gender of the Vaccination Attitudes Examination (VAX) Scale in Colombian University Students"

_jcm, 2022, doi:10.3390/jcm11164682_

Round 1

Reviewer 1 Report

Manuscript jcm-1802771, entitled "Evidence of validity and measurement invariance by gender of the Vaccination Attitudes Examination (VAX) scale in Colombian university students", is an interesting article investigating the psychometric properties of the VAX scale in a sample of 1,074 participants. The article is based on data collected between August 2021 and April 2022, as a part of an international multidisciplinary project on the impact of Covid-19.

The rationale of the study is well described, with appropriate relevant literature references. The study collected sociodemographic data, the VAX scale data, and the Covid-19 vaccination status. The study methodology is adequate, with a solid statistical analysis, and the results are clearly presented. The discussion is well balanced and adequately supported by the data. The quality of written English is suitable for publication.

In my opinion, this article is acceptable for publication after minor revision. The following issues need to be addressed to make it suitable for publication in Journal of Clinical Medicine.

Major Compulsory Revision

1. Methods. Although this study is a part of a project on the impact of Covid-19, it is largely oriented towards validation of the VAX scale, which is about vaccination in general. The only link between the VAX scale data and the Covid-19 vaccination is the data collection period (during the pandemic).

It would have been desirable, to better understand the reasons for the non-vaccination against Covid-19, to have additional information, for example about having been infected or not by the virus. If this information is available in the study, the authors could take it into account in a sensitivity analysis, by considering the infection as equivalent to a vaccine dose, or by carrying out the concurrent validity analysis on the sub-population of people who have never been infected. On such a population, we expect a better predictive ability of the VAX scale with respect to the Covid-19 vaccination.

Minor Compulsory Revisions

2. Results and Table 1. Descriptive results should include other available information, for example age distribution of the study population and especially Covid-19 vaccination status, and according to the number of doses.

3. Appendix. It would have been useful to have also the English version of the 12 items of the VAX scale for a better understanding.

Author Response

Manuscript jcm-1802771, entitled "Evidence of validity and measurement invariance by gender of the Vaccination Attitudes Examination (VAX) scale in Colombian university students", is an interesting article investigating the psychometric properties of the VAX scale in a sample of 1,074 participants. The article is based on data collected between August 2021 and April 2022, as a part of an international multidisciplinary project on the impact of Covid-19.

The rationale of the study is well described, with appropriate relevant literature references. The study collected sociodemographic data, the VAX scale data, and the Covid-19 vaccination status. The study methodology is adequate, with a solid statistical analysis, and the results are clearly presented. The discussion is well balanced and adequately supported by the data. The quality of written English is suitable for publication.

In my opinion, this article is acceptable for publication after minor revision. The following issues need to be addressed to make it suitable for publication in Journal of Clinical Medicine.

Answer. Thank you very much for your comments.

Major Compulsory Revision

Methods. Although this study is a part of a project on the impact of Covid-19, it is largely oriented towards validation of the VAX scale, which is about vaccination in general. The only link between the VAX scale data and the Covid-19 vaccination is the data collection period (during the pandemic).

Answer. Thank you very much for this comment. The validation of this questionnaire has been part of a broader project in which, among other things, the aim is to study the psychological well-being of university students in times of pandemic. For this reason, participants were only asked about vaccination against the coronavirus and not if they had received other vaccines.

It would have been desirable, to better understand the reasons for the non-vaccination against Covid-19, to have additional information, for example about having been infected or not by the virus. If this information is available in the study, the authors could take it into account in a sensitivity analysis, by considering the infection as equivalent to a vaccine dose, or by carrying out the concurrent validity analysis on the sub-population of people who have never been infected. On such a population, we expect a better predictive ability of the VAX scale with respect to the Covid-19 vaccination.

Answer. Thank you very much for this interesting comment. We do have information on the people who report having been infected. Specifically, it was asked if the participants had been infected with COVID and there were three possible answers: yes (29.5%), no (46.4%), not sure (20.6%).

According to the reviewer's suggestion, we have estimated the validity model with the sample of people who say they have not been infected or are not sure about it. The biggest difference is that statistically significant correlations are observed between all the factors, something that does not occur with the validity model in the manuscript. In this model (with all the sample), the correlation between factors 1 (trust of vaccine benefit) and 2 (worries over unforeseen future effects), and the correlation between factors 1 (trust of vaccine benefit) and 4 (preference for natural immunity) was not significant. However, when performing the analysis after eliminating the infected people, the correlation between these factors is significant and inverse. Despite this, this result does not bring about major changes in the validity model, in which factor 1 (trust of vaccine benefit) and factor 2 (worries over unforeseen future effects) also continue to be significant predictors, and the odds ratio are also very similar.

If we further reduce the sample and estimate the model only with the people who say they have not been infected, again the prediction model gives the same results as the one in the manuscript, except for the fact that only the correlation between factors 1 (trust of vaccine benefit) and 4 (preference for natural immunity) is not significant. However, factors 1 (trust of vaccine benefit) and 2 (worries over unforeseen future effects) also predict vaccination or not, and the odds ratio are very similar.

Although the new models do not provide relevant changes with respect to the original model, we have included this information in the manuscript (lines 272-290). We have also provided information regarding people infected and vaccinated in Table 1 as part of the sample description.

Minor Compulsory Revisions

Results and Table 1. Descriptive results should include other available information, for example age distribution of the study population and especially Covid-19 vaccination status, and according to the number of doses.

Answer. Thank you very much for this comment. As we stated before, we have also provided this information in Table 1 as part of the sample description.

Appendix. It would have been useful to have also the English version of the 12 items of the VAX scale for a better understanding.

Answer. Thank you very much for this comment. Items in English are available in the original article, and which is cited in the manuscript. For this reason, we consider that it is not necessary to include them, and we have only provided the Spanish version.

Reviewer 2 Report

The manuscript is poorly written and the data is not enough to be published as full article. Also, all the statistical analyses need to be repeated. Also, figures are confused and the authors did not describe them properly within the text.

Author Response

Thank you very much for your valuable comments. We will answer each question separately.

  1. Regarding the sample used in this study, in lines 112 to 118 we explain why we conducted the study with university students. Additionally, in the procedure section (lines 134 to 138), it is explained that this sample comes from a larger study in which it is intended to study psychological well-being in university students. We consider your recommendation to study the psychometric properties of the VAX in other types of samples in future studies (lines 357-362).

  1. Thank you very much for your suggestion. Regarding the analyses, we have not only carried out a confirmatory factorial analysis. We have also studied the concurrent validity using structural equation modeling, as well as the measurement invariance by gender (lines 22-23).

  1. Thank you very much for your support. We have modified this sentence in lines 26 to 28.

  1. Thank you very much for this question. This statement is the result of the review of the study published by the World Health Organization. For clarity, we have included the citation again on line 54.

  1. Thank you very much for your suggestion. Usually in the psychometric context the term "instrument" is used. However, following the reviewer's suggestion, we have replaced the term "instrument" with the term "tool" (line 86, abstract and conclusions).

  1. Thank you very much for this question. Regarding the sample of university students, they are not students at a single university. Table 1 specifies the centers where some of the participants studied.

  1. Thank you very much for this comment. Regarding people who do not believe in COVID, that question has not been asked, so we do not have that information. It is interesting to consider it in future studies. We include it in the Discussion (lines 360-362). Additionally, we added the validity model with the sample of people who say they have not been infected or are not sure about it, and the new models do not provide relevant changes with respect to the original model (lines 274-292).

  1. Thank you very much for this comment. Regarding the accuracy of the online survey, it is true that in all types of surveys (online or paper and pencil) it can happen that answers are falsified or that social desirability appears. However, as this survey is anonymous and voluntary, we believe that these cases, if they exist, must be very few. In addition, the participants were informed that at any time they could delete their responses and exit the survey. We believe that this type of facility allows people to respond more calmly, since they feel that they are always in control of their responses, and therefore unreliable data will be very few. Even so, the data file has been refined and it has been verified that there are no inconsistent responses (for example ages outside the possible range), or the same response in all the items of the same questionnaire.

  1. Thank you very much for this comment. Regarding the comparison suggested by the reviewer with data from other countries, we have not done it because it is not the objective of this study. Our goal has been to validate the VAX in Colombia and study its psychometric properties. Later we do want to carry out studies with this questionnaire, putting it in relation to other variables.

  1. Thank you very much for this question. The response effect of the different social networks has not been evaluated; it was not the objective of this study. They have been used only to access more students and be able to collect more data. The social networks through which the link has been distributed have been those of the collaborators in the study.

  1. Thank you very much for this question. Regarding the size of the sample, many authors (for example, Kyriazos (2018), Sim et al. (2022), and Ferrando et al. (2022) would consider that this sample is adequate to be able to factorize the 15 items of the scale, as well as to study its concurrent validity. In addition, as mentioned in the manuscript, the objective of this study is part of a broader project that aims to analyze the psychological well-being of Colombian university students (lines 132 to 134), so access to university employees has not been considered.

  1. Thank you very much for your appreciation. The different socioeconomic levels are clearly established by the Colombian government depending on the services offered in the area where people live. This variable has been included for future studies and as an important part of the description of the sample. The same happens with the degree that the participants are studying. We consider it interesting to know this variable for future studies.

  1. Thank you very much for this question. Face-to-face does not refer to how the survey data was collected, but to how students are doing in their studies. At the time the survey data was collected, some students were studying in person, others online, and others combined (both). We consider it interesting to provide this information, which could also be interesting in future studies.

  1. We thank you very much for your appreciation, although we consider that the results are correctly described. If the reviewer can make any suggestions for improvement in the writing, we are willing to consider it.

  1. Thank you very much for this appreciation. We have carefully revised the Discussion and have redrafted some paragraphs (lines 304-307). Additionally, we have commented on our results by comparing them with those of previous versions of the VAX because we believe that it is one of the contributions of our research. The novelty of the study is that the measurement invariance by gender has been studied, which has not been done in any of the previous adaptations of the VAX, and we draw attention to this in the discussion (lines 323 to 343).

  1. Thank you very much for this comment. We have included the previously mentioned limitations, although we consider that they do not affect the results obtained in this study (lines 357-362).

  1. Thank you very much for your suggestion. The conclusions have been rewritten (lines 375-382).

Kyriazos, T. A. (2018). Applied Psychometrics: Sample Size and Sample Power Considerations in Factor Analysis (EFA, CFA) and SEM in General. Psychology, 9, 2207-2230. https://doi.org/10.4236/psych.2018.98126

Sim, M., Kim, S.-Y., & Suh, Y. (2022). Sample Size Requirements for Simple and Complex Mediation Models. Educational and Psychological Measurement, 82(1), 76–106. https://doi.org/10.1177/00131644211003261

Ferrando-Piera, P. J., Lorenzo-Seva, U., Hernández-Dorado, A., & Muñiz-Fernández, J. (2022). Decálogo para el Análisis Factorial de los Ítems de un Test [Decalogue for the Factor Analysis of Test Items]. Psicothema, 34(1), 7-17. https://doi.org/10.7334/psicothema2021.456

Round 2

Reviewer 2 Report

Thanks for responding to my comments